# CONTEXTGNN: BEYOND TWO-TOWER RECOMMENDATION SYSTEMS

**Yiwen Yuan, Zecheng Zhang, Xinwei He, Akihiro Nitta,**
**Weihua Hu, Manan Shah, Blaž Stojanovič, Shenyang Huang,**
**Jan Eric Lenssen, Jure Leskovec, Matthias Fey**
Kumo.AI

## ABSTRACT

Recommendation systems predominantly utilize two-tower architectures, which evaluate user-item rankings through the inner product of their respective embeddings. However, one key limitation of two-tower models is that they learn a pair-agnostic representation of users and items. In contrast, pair-wise representations either scale poorly due to their quadratic complexity or are too restrictive on the candidate pairs to rank. To address these issues, we introduce *Context-based Graph Neural Networks* (CONTEXTGNNs), a novel deep learning architecture for link prediction in recommendation systems. The method employs a pair-wise representation technique for familiar items situated within a user's local subgraph, while leveraging two-tower representations to facilitate the recommendation of exploratory items. A final network then predicts how to fuse both pair-wise and two-tower recommendations into a single ranking of items. We demonstrate that CONTEXTGNN is able to adapt to different data characteristics and outperforms existing methods, both traditional and GNN-based, on a diverse set of practical recommendation tasks, improving performance by 20% on average.

## 1 INTRODUCTION

Recommendation systems have emerged as an important application domain for predictive machine learning over the past decades (Webber, 2021; He et al., 2023; Li et al., 2024). Given a set of users and a set of items (*e.g.*, products available for purchase), recommendation systems aim to identify optimal item recommendations for each user (*e.g.*, products most likely to be purchased by the user). Traditionally, this problem is modeled via different variants of a *two-tower* paradigm (Hu et al., 2008; Koren et al., 2009), where one tower embeds users and the other tower embeds items, which are then matched and ranked via an inner-product decoder. This scheme proves to be highly efficient for scaling up recommendation systems during the inference phase, as it allows to pre-compute user and item representations and to perform the final ranking via fast (approximate) *maximum inner product search* (Johnson et al., 2019).

However, one key limitation of two-tower based architectures for recommendation is that they learn a *pair-agnostic* representation for users and items. That is, the user representation is not aware of the item under consideration, and similarly, the item representation is not aware of the user and thus item representations are not capturing the uniqueness of user's view on the items. As such, neither of the representations on both ends capture knowledge about the pair-wise dependency they are making a prediction for. For example, consider a user who restocks their cosmetic products on a regular basis. In this scenario, the fine-grained context of user-cosmetic pairs is crucial, which cannot be adequately captured by two independent user and item representations alone. Such lack of knowledge has severe consequences on the quality of predictions, since, *e.g.*, the model is unable to distinguish between scenarios such as familiar purchases (*i.e.* users who repeatedly interact with the similar set of items) *vs.* exploratory purchases (*i.e.* users who like to explore new items). While pair-wise representations, which incorporate the knowledge about the pair they are making a prediction for, are able to *contextualize* the prediction, one would need to generate pair-wise representations for all possible user-item pairs, which is ineffable due to its quadratic complexity. Alternatively, one can pre-filter the set of candidate pairs, *e.g.*, via content-based filtering or collaborative filtering Ricci et al. (2010); Campana & Delmastro (2017), but the model's capabilities are then limited by the recall of the candidate generation procedure.

Here, we develop *Context-based Graph Neural Networks* (CONTEXTGNNs), a novel single-stage *Graph Neural Network (GNN)*-based recommendation system that fuses pair-wise representations and two-tower representations into a single unified architecture. CONTEXTGNN is designed to perform *temporal* recommendations (predict the next set of items a user will interact with) on a *heterogeneous temporal* graph, and leverages the best of both worlds: The first model supports pair-wise representations for all items *within* a user's *local* interaction graph (Zhu et al., 2021). For *distant* user-item pairs, *i.e.* user-item pairs that are not within the $k$-hop neighborhood of each other, the pair-wise embeddings are supplemented by a *global*, two-tower-based GNN, which serves as an effective fallback model. A final network then predicts a user-specific, personalized fusion score that dictates how to merge both pair-wise and two-tower recommendations into a single ranking.

They key idea behind CONTEXTGNN is to contextualize the prediction for the area of items for which a user has rich past interactions. Here, pair-wise representations are able to capture fine-grained patterns of past user-item interactions, such as repeat purchases and other prior actions (*e.g.*, clicks, tag-as-favorite, add-to-cart), or through collaborative signals (*e.g.*, whether friends have purchased the same item). The second component, a novel pair-agnostic two-tower model based on shallow item embeddings, allows for modeling exploratory and serendipitous recommendations. For these "distant" items, where no specific user-item context exists, pair-wise representations offer minimal benefits, making the fallback to two-tower representations suitable. Importantly, the two-tower model is designed in such a way that it allows for integration into pair-wise architectures with little computational overhead. Finally, the model learns which users prefer familiar items over exploratory purchases and adjusts the final ranking scores accordingly.

We deploy CONTEXTGNNs in the context of relational deep learning (Fey et al., 2024) for recommendation within relational databases. Relational databases (Robinson et al., 2024) comprise diverse sets of tables with rich multi-modal input features and rich multi-behavioral, temporal interactions, making them the perfect setting for stress-testing our model. Furthermore, we examine the challenges in modeling the complex patterns of human behavior. Our analysis and experimental results underscore the need for a hybrid architecture, as a single model is not sufficient to address the diversity of real-world datasets both within and across tasks effectively. We demonstrate that CONTEXTGNN is able to adapt to different data characteristics, while outperforming existing models on realistic and practical recommendation tasks. In particular, CONTEXTGNN improves results by 20% on average compared to the best pair-wise representation baseline, and by 344% on average compared to the best two-tower representation baseline.

## 2 RELATED WORK

Recommendation systems are a long-standing research area in machine learning. The recommendation problem is often formulated as *link prediction* on a *bipartite graph* with two sets of nodes, one containing the users and another containing the items. Between these two node sets, past interactions are presented as links, and the goal is to predict which links are going to occur in the future. Traditionally, it is approached by computing the inner product of user and item embeddings, *e.g.*, via matrix factorization (Koren et al., 2009) to discover and utilize similarity between users and items, *i.e.* collaborative filtering (Hu et al., 2008; Mnih & Salakhutdinov, 2007).

Similar to many other areas, deep learning made an impact in recommendation systems, *e.g.*, by predicting scores with non-linear networks (H. & C., 2017; He et al., 2017), by utilizing graph embedding techniques (Perozzi et al., 2014; Grover & Leskovec, 2016), or by sampling from distributions modeled by a variational autoencoder (Liang et al., 2018). Recently, the new field of generative recommendation deserves mentioning, utilizing large language models to provide recommendation by text generation (Wu et al., 2023). Our work here relates to two lines of research: Sequential recommendation systems and recommendations with GNNs.

**Recommendation on Sequences.** Recommendation has traditionally been modeled as a sequence prediction problem. Early methods utilize Marcov chains (Rendle et al., 2010; He & McAuley, 2016), which have later been replaced by recurrent neural networks (Liu et al., 2018), temporal attention (Kang & McAuley, 2018; Li et al., 2017) and transformer formulations (Sun et al., 2019; Yang et al., 2022; Xia et al., 2022). They also have been combined with GNNs to additionally capture collaborative signals (Ma et al., 2020).

**Recommendation with GNNs.** Various two-tower-based Graph Neural Networks (Hamilton et al., 2017) have been developed for link prediction tasks, based on the idea of integrating the paradigm of collaborative filtering via refining embeddings on the user-item interaction graph. Early examples use GNN-based autoencoders (Kipf & Welling, 2016; Schlichtkrull et al., 2018) to obtain prediction on graph links. Since GNNs propagate information locally (and thus cannot reason about the position of nodes inside a graph), it has been shown that propagating shallow user and item embeddings with deep GNNs is crucial to capture the full collaborative signal (Wang et al., 2019). LIGHTGCN (He et al., 2020) and ULTRAGCN (Mao et al., 2021) further simplify message propagation for embedding generation for improved performance on small-scale datasets. Another line of work strengthens GNN-based recommender systems by adding self-supervision (Wu et al., 2021; Yu et al., 2022; Cai et al., 2023), *e.g.*, based on contrastive learning via graph augmentations. These findings are orthogonal to our work, but still embrace a two-tower architecture under the hood.

The line of work that comes closest to CONTEXTGNN is related to encoding identity- and position-awareness into GNNs in order to relate user-item pairs (You et al., 2019; 2021). For example, local subgraphs have been extended to encode positional information of user-item pairs (Zhang & Chen, 2018; Teru et al., 2020), but lack an efficient inference procedure. NBFNET (Zhu et al., 2021) introduces a path-based pair-wise representation model that constructs a subgraph centered around the user node, while reading out the GNN's representations at the item nodes. This method yields item representations that are conditioned on the specific user, thereby incorporating pair-wise information without incurring excessive computational cost. However, this method constrains candidates to items within the user's subgraph, which tremendously limits its effectiveness in scenarios involving exploratory recommendations or cold-start items. Our local pair-wise representation model builds upon this framework, extending it to fit into the heterogeneous, multi-behavioral, and temporal recommendation system context. Most importantly, we analyze and eliminate its main limitation around locality by extending it with an effective fallback model.

While the vast majority of related work focuses on modeling recommendation systems on static graphs, recently multiple benchmarks for temporal recommendation have appeared, such as the 🐢GB TEMPORAL GRAPH BENCHMARK (Huang et al., 2023) and 🔧 RELBENCH (Robinson et al., 2024). In this work, we adapt the temporal formulation of a recommender systems task and extensively evaluate CONTEXTGNN in the practical setting of 🔧 RELBENCH and show that our method significantly outperforms previous approaches.

## 3 TEMPORAL RECOMMENDATIONS AND WHERE TO FIND THEM

We formulate the temporal recommendation problem on a *heterogeneous graph snapshot* $\mathcal{G}^{(-\infty, T]} = (\mathcal{V}, \mathcal{E}, \phi, \psi)$ up to timestamp $T$, with node set $\mathcal{V}$ and edge set $\mathcal{E} \subseteq \mathcal{V} \times \mathcal{V}$, where each node $v \in \mathcal{V}$ belongs to a *node type* $\phi(v)$ and each edge $e \in \mathcal{E}$ belongs to an *edge type* $\psi(e)$. We refer to $\mathcal{L} \subset \mathcal{V}$ as the user set and $\mathcal{R} \subset \mathcal{V}$ as the item set within our heterogeneous graph. The task is then to predict a set of ground-truth items $\mathcal{Y}_v^{(T, T+i]} \subseteq \mathcal{R}$, for which there occurred a link for the user $v \in \mathcal{L}$ in the time interval $(T, T + i]$ for a given interval size $i$. When making the prediction, the model only has access to historical information up to timestamp $T$.

In the simplest case, $\mathcal{G}^{(-\infty, T]}$ is given as a bipartite graph, with node set $\mathcal{V} = \mathcal{L} \cup \mathcal{R}$, and a single edge type $|\{\psi(e) : e \in \mathcal{E}\}| = 1$. However, the generalization to heterogeneous graphs allows to incorporate other node types, but most importantly, different edge types of user behaviors (*e.g.*, click, tag-as-favorite, add-to-cart) which offer complementary signals for predicting the target behavior (*e.g.*, purchase) (Xia et al., 2022). Nodes and edges may be (partially) annotated with an initial feature representation, *e.g.* $\boldsymbol{h}_v^{(0)} \in \mathbb{R}^d, v \in \mathcal{V}$, and with a timestamp indicating their appearance (or disappearance). In the absence of timestamps, the given framework can be easily transformed into a *static* link prediction problem, as commonly practiced in literature (Kipf & Welling, 2016; Wang et al., 2019; He et al., 2020). However, this approach disregards critical information, such as the recency of past activity. Note that the given framework also allows newly appearing users and items over time. However, for simplicity of notation, we assume them to be in $\mathcal{L}$ and $\mathcal{R}$ at all times.

**Locality Score.** Developing machine learning solutions for recommendation systems is inherently challenging due to the complex patterns of human behavior (He et al., 2023). Users vary significantly in preferences; some are explorers who constantly seek new experiences, while others are repeaters

who prefer familiarity. Such characteristics are especially noticeable across different datasets, *e.g.*, it is more likely that a user repeats a previous purchase than that the user rates the same movie twice.

To understand these effects better, we introduce the *locality score*, which measures the fraction of ground-truth links that fall within the $k$-hop neighborhood $\mathcal{N}_k^{(-\infty, T]}(v)$ of the subgraph centered around a user $v \in \mathcal{L}$ up to timestamp $T$:

$$s_k^{(T, T+i]} = \frac{1}{|\mathcal{L}|} \sum_{v \in \mathcal{L}} = \frac{\left| \mathcal{N}_k^{(-\infty, T]}(v) \cap \mathcal{R} \cap \mathcal{Y}_v^{[T, T+i]} \right|}{\left| \mathcal{Y}_v^{[T, T+i]} \right|}. \tag{1}$$

Intuitively, the locality score quantifies how many of its future ground-truth items are local to a user. Depending on the subgraph depth $k$, the score reflects different interaction patterns. For a shallow subgraph ($k = 1$), it captures whether the user repeats purchases or has previously interacted with the item through actions like clicks or tagging as favorite. As the subgraph depth increases, the score begins to incorporate collaborative filtering signals, such as whether friends have purchased the ground-truth item in the past ($k = 2$), or whether the ground-truth item was previously purchased by users that have a similar buying behavior as the user under consideration ($k = 3$). Note that in practice, it is infeasible to go beyond depth $k = 3$ due to scalability concerns.

Table 1: The **locality score** for different subgraph depths $k \in \{1, 3\}$ on validation/test splits for all recommendation tasks in RELBENCH.

| Dataset | Task | Split | $k = 1$ | $k = 3$ |
|---------|------|-------|---------|---------|
| **rel-amazon** | user-item-purchase | val | 0.002 | 0.181 |
| | | test | 0.001 | 0.168 |
| | user-item-rate | val | 0.002 | 0.183 |
| | | test | 0.002 | 0.170 |
| | user-item-review | val | 0.002 | 0.185 |
| | | test | 0.001 | 0.175 |
| **rel-hm** | user-item-purchase | val | 0.045 | 0.395 |
| | | test | 0.048 | 0.417 |
| **rel-stack** | user-post-comment | val | 0.265 | 0.308 |
| | | test | 0.256 | 0.298 |
| | post-post-related | val | 0.109 | 0.225 |
| | | test | 0.141 | 0.280 |
| **rel-trial** | condition-sponsor-run | val | 0.000 | 0.241 |
| | | test | 0.000 | 0.242 |
| | site-sponsor-run | val | 0.000 | 0.208 |
| | | test | 0.000 | 0.229 |

We evaluate the locality score for varying $k$ on RELBENCH (Robinson et al., 2024), a relational deep learning (Fey et al., 2024) benchmark that spans a wide variety of real-world recommendation tasks with diverse characteristics (*cf.* Table 1). Interestingly, the locality score does not exceed 0.5 on any dataset, indicating that the majority of ground-truth items are entirely *exploratory* and not covered by any path of length $\leq 3$ in the graph. However, for $k = 1$, we observe high locality scores on rel-hm and rel-stack, suggesting that users tend to frequently repeat clothing purchases or comment multiple times on the same post. Increasing the subgraph depth generally improves coverage as seen in datasets like rel-amazon. While users do not usually repurchase/rate/review the same item twice here, they tend to purchase the same products that "nearby" users bought. Overall, this observation suggests that while pair-wise representation architectures such as NBFNET (Zhu et al., 2021) are powerful, they often times fail to offer sufficient coverage within their candidate set, decreasing their effectiveness tremendously. This raises an interesting question of how we can leverage the benefits of pair-wise representations, while supporting the modeling of exploratory recommendations without the need for complex multi-stage approaches.

## 4 RECOMMENDATION WITH CONTEXT-BASED GRAPH NEURAL NETWORKS

Next, we describe *Context-based Graph Neural Networks* (CONTEXTGNNs), which consist of two separate GNN architectures sitting behind the same GNN backbone. CONTEXTGNN fuses both pair-wise representations and two-tower representations into a single architecture, and is thus naturally able to adapt to diverse dataset and task characteristics. The first model employs pair-wise representations based on the item candidate set given within the local user-centric subgraph (*cf.* Sec. 4.1). The second model introduces a novel two-tower architecture based on shallow item representations, which is used to predict a ranking for all pairs of users and items outside the user's subgraph (*cf.* Sec. 4.2). Finally, a user-specific fusion score, produced by an MLP on the user GNN representation, is added to the scores of the pair-wise representation model, which aligns the distinct

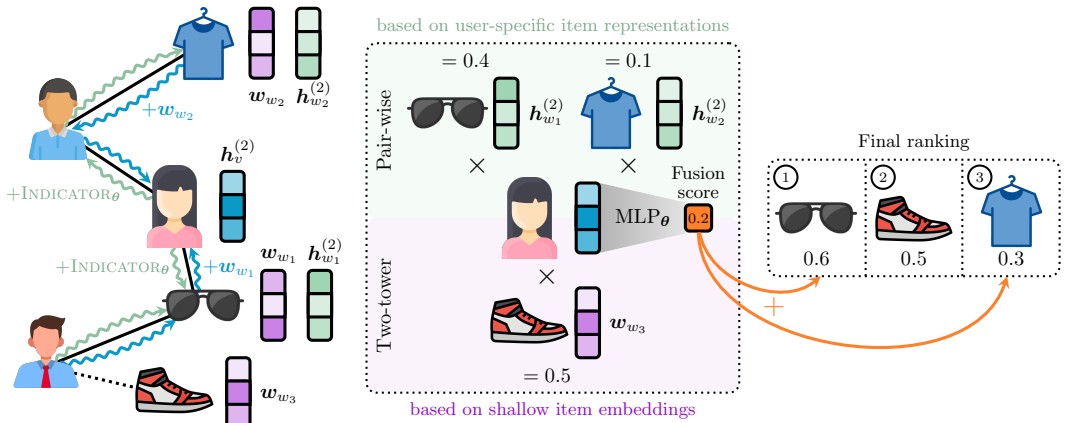

Figure 1: **Overview of Context-based Graph Neural Networks.** CONTEXTGNN utilizes a bidirectional $\text{GNN}_{\boldsymbol{\theta}}$ to learn user $\boldsymbol{h}_v^{(2)}$ and user-specific item representations $\boldsymbol{h}_w^{(2)}$ within a user's local subgraph. Its message passing scheme is enhanced by additionally propagating shallow item embeddings $\boldsymbol{w}_w$ and seed user $\text{IDENTICATOR}_{\boldsymbol{\theta}}$ representations. Afterwards, item scores are produced depending on whether an item situates within a user's subgraph. A user-specific fusion score is learned via an $\text{MLP}_{\boldsymbol{\theta}}$ to produce the final ranking by offsetting the contributions of local rankings.

scores of the two models and captures *how* exploratory a specific user is, thus given either more or less weight to the respective models (*cf.* Sec. 4.3). We go over additional considerations and extensions in Sec. 4.4. An overview of CONTEXTGNN is visualized in Fig. 1.

## 4.1 PAIR-WISE REPRESENTATIONS

Our local pair-wise representation model builds upon the framework proposed by Zhu et al. (2021), extending it to fit into the heterogeneous, multi-behavioral, and temporal recommendation system context. NBFNET is a path-based method that generalizes the Bellman-Ford algorithm (Baras & Theodorakopoulos, 2010). Rather than learning a pair-wise representation via two independent node representations $\boldsymbol{h}_v^{(k)}$ and $\boldsymbol{h}_w^{(k)}$ (*i.e.* via two independent subgraphs), NBFNET only utilizes the user-specific subgraph, but readouts GNN item representations from it. As such, NBFNET abbreviates the pair-wise representation $\boldsymbol{h}_{(v,w)}^{(k)}$ via the user-specific item representations $\boldsymbol{h}_w^{(k)}$. Hence, it captures both knowledge about the item $w$ and the user $v$ in its representation. In order to let the GNN differentiate between the seed user and other users being sampled within the subgraph, a $d$-dimensional $\text{INDICATOR}_{\boldsymbol{\theta}}$ representation is added to the seed node (You et al., 2021). Specifically, our pair-wise representation model then looks as follows:

1. Sample a $k$-hop subgraph $\tilde{\mathcal{G}} \leftarrow \mathcal{G}_k^{(-\infty, T]}[v]$ with node set $\tilde{\mathcal{V}}$ around user $v \in \mathcal{L}$.

2. Add an indicator representation to the user seed node: $\boldsymbol{h}_v^{(0)} \leftarrow \boldsymbol{h}_v^{(0)} + \text{INDICATOR}_{\boldsymbol{\theta}}$.

3. Read out both GNN user and item representations at layer $k$:

$$\boldsymbol{h}_v^{(k)}, \{\boldsymbol{h}_w^{(k)} : w \in \tilde{\mathcal{V}} \cap \mathcal{R}\} \leftarrow \text{GNN}_{\boldsymbol{\theta}}^{(k)}(\tilde{\mathcal{G}}, \boldsymbol{H}^{(0)}).$$

4. Compute the final ranking for all items $w \in \tilde{\mathcal{V}} \cap \mathcal{R}$: $y_{(v,w)}^{(\text{pair})} \leftarrow \boldsymbol{h}_v^{(k)} \cdot \boldsymbol{h}_w^{(k)}$.

In contrast to NBFNET, we readout both the user and item representations from the GNN in order to produce the ranking of local items. This lets the model leverage the full $k$-hop user information, as otherwise user representations are only computed in intermediate GNN layers based on limited subgraph depths.

Further note that sampling a $k$-hop subgraph typically results in a *directed* computation graph towards the seed node (Fey & Lenssen, 2019). However, to facilitate the extraction of item node representations, the sampled subgraph must be transformed into a bidirectional graph prior to applying the GNN. This approach aligns with previous work that decouples the depth and scope of GNNs (Zeng et al., 2021).

This approach proves particularly effective in the context of temporal, heterogeneous graphs, as the GNN naturally learns to integrate multi-behavior signals originating from different edge types. For instance, the pair-wise representation captures all past user-item interactions, such as the recency of the last purchase or whether the item has been previously clicked or tagged. Despite its expressiveness, this method is also highly efficient, as only a single GNN pass over the user subgraph is required to make predictions for the entire set of related items. Assuming bounded subgraph sizes, training and inference over the full user set can be achieved in $\mathcal{O}(|\mathcal{L}|)$ time, which is in stark contrast to the $\mathcal{O}(|\mathcal{L}| \cdot |\mathcal{R}|)$ complexity required by two-tower models. Moreover, as no shallow embeddings are used, this approach naturally extends to newly appearing users and items over time. Nonetheless, as discussed in Sec. 3, this method alone does not fully address the diverse requirements of modern recommendation systems due to its limited set of potential candidates.

## 4.2 TWO-TOWER REPRESENTATIONS

CONTEXTGNN's two-tower model ranks all user-item pairs outside the user's subgraph, serving as an effective fallback mechanism to supplement the pair-wise representations. While drawing inspiration from related two-tower architectures (Wang et al., 2019; He et al., 2020), our novel two-tower model enables an efficient integration with pair-wise architectures. The key innovation in our two-tower model is the use of *shallow* item representations. Specifically, we do *not* deploy a GNN to compute item representations; instead we entirely rely on a shallow embedding matrix $\boldsymbol{W} \in \mathbb{R}^{|\mathcal{R}| \times d}$ to learn item representations. This design is inspired by multiple key observations:

- **Limited information gain from applying a GNN on the item side.** Item connections are naturally *very* dense. In extreme cases, popular items may receive over 1M interactions (an item is connected to every user that has previously interacted with it). This leads to significant challenges and uncertainties in subgraph sampling, and easily exposes the GNN to oversquashing and oversmoothing issues.
- **Shallow embedding matrices are very effective.** Shallow item embedding matrices can capture key signals such as popularity trends, seasonal patterns, demographic preferences and item similarity just as effectively as a GNN.
- **GNN item representations scale poorly during training.** Incorporating both user and item GNN representations limits the model to only use a small number of negative samples due to memory constraints. In contrast, shallow item embeddings support training against a much larger corpus of negative samples ($\approx$ 10M), which is critical for improving model performance. This approach also eliminates the need for complex negative sampling strategies, which are often required when the number of negative samples is limited.

Importantly, we also inject the shallow item embeddings within the user's GNN forward pass, such that user representations can better align themselves to the corresponding item representations. Hence, the GNN representation of users follows a similar spirit to models such as NGCF (Wang et al., 2019) or LIGHTGCN (He et al., 2020), although we do not utilize shallow user embeddings as the GNN itself is powerful enough to learn its own rich representation without them. As such, our two-tower representation model can be summarized as follows:

1. Sample a $k$-hop subgraph $\tilde{\mathcal{G}} \leftarrow \mathcal{G}_k^{(-\infty, T]}[v]$ with node set $\tilde{\mathcal{V}}$ around user $v \in \mathcal{L}$.

2. Add the shallow embedding to all sampled items $w \in \tilde{\mathcal{V}} \cap \mathcal{R}$: $\boldsymbol{h}_w^{(0)} \leftarrow \boldsymbol{h}_w^{(0)} + \boldsymbol{w}_w$.

3. Read out the GNN user representation at layer $k$: $\boldsymbol{h}_v^{(k)} \leftarrow \text{GNN}_{\boldsymbol{\theta}}^{(k)}(\tilde{\mathcal{G}}, \boldsymbol{H}^{(0)})$.

4. Compute the final ranking for all items $w \in \mathcal{R} \setminus \tilde{\mathcal{V}}$: $y_{(v,w)}^{(\text{tower})} \leftarrow \boldsymbol{h}_v^{(k)} \cdot \boldsymbol{w}_w$.

## 4.3 CONTEXT-BASED GRAPH NEURAL NETWORKS

Our final CONTEXTGNN architecture fuses both pair-wise representations and two-tower representations into a single unified architecture. That is, for all items $w \in \tilde{\mathcal{V}} \cup \mathcal{R}$ *inside* the local user subgraph, we leverage the scores $y_{(v,w)}^{(\text{pair})}$ obtained from the pair-wise representations, while we fallback to the two-tower scores $y_{(v,w)}^{(\text{tower})}$ for all items $w \in \tilde{\mathcal{V}} \setminus \mathcal{R}$ *outside* the sampled subgraph.

The observation that user behaviors are diverse is the final element that makes CONTEXTGNN work. To accommodate such diversity across different users, CONTEXTGNN learns a user-specific *fusion score* predicted from the GNN's user embeddings $h_v^{(k)}$ via an MLP$_\theta$. This personalized fusion score aligns the distinct scores by learning which users prefer familiar items over exploratory purchases, and adjusts the final ranking scores accordingly. With this, the final score is given by

$$y_{(v,w)} = \begin{cases} y_{(v,w)}^{(\text{pair})} + \text{MLP}_{\theta}\left(h_v^{(k)}\right) & \text{if } w \in \tilde{\mathcal{V}} \cup \mathcal{R}, \\ y_{(v,w)}^{(\text{tower})} & \text{otherwise.} \end{cases} \tag{2}$$

In CONTEXTGNN, both pair-wise representations and two-tower representations are derived from the *same* GNN backbone. Since both models depend solely on the user subgraph, the user embedding $h_v^{(k)}$ can be extracted in a single GNN forward pass and leveraged for downstream uses in both models. This streamlined approach makes CONTEXTGNN computationally very efficient. CONTEXTGNN is then trained end-to-end, optimizing both types of item scores as well as the fusion score altogether to maximize the predictive performance of future user-item interactions. In practice, we utilize the cross entropy loss for optimization, although any other loss formulation is applicable as well (Rendle et al., 2009). During inference, we obtain top scores from the two-tower model via (approximate) *maximum inner product search* (Johnson et al., 2019), and merge them to the pair-wise scores in a post-processing procedure.

## 4.4 EXTENSIONS

We go over additional considerations and extensions when applying CONTEXTGNNs in practice.

**Fitting into the Context of Relational Deep Learning.** CONTEXTGNNs nicely align with the framework of relational deep learning (Fey et al., 2024), a blueprint for graph representation learning on relational databases. Specifically, relational deep learning treats relational databases as a heterogeneous temporal graph, in which dimension and fact tables are linked through primary-foreign key relationships. Then, temporal-aware subgraph sampling is employed to generate heterogeneous subgraph snapshots $\mathcal{G}_k^{(-\infty,T]}[v]$ up to timestamp $T$, which are used to predict future user-item interactions. Relational databases (Robinson et al., 2024) comprise diverse set of tables with rich multi-modal input features and rich multi-behavioral, temporal interactions, making them the perfect setting for stress-testing CONTEXTGNNs.

**Transductive *vs.* Inductive Modeling.** Although the pair-wise representations in CONTEXTGNN are inductive by design, the usage of the shallow embedding matrix on the item side places it in a transductive setting. This means that while CONTEXTGNN can naturally accommodate newly appearing users over time, it is unable to handle new items at prediction time since their shallow embeddings remain uninitialized. To enable CONTEXTGNN to operate in an inductive setting, we propose to replace the shallow item embedding matrix with a deep neural network on top of its item input features $h_w^{(0)}$. This approach allows CONTEXTGNN to scale to tasks such as marketplace or event recommendation, where handling of new items is essential.

**Sampled Softmax Formulation.** Since it is infeasible to train against the full item set $\mathcal{R}$ when optimizing CONTEXTGNNs, in practice, we rely on a sampled softmax formulation. This basically converts the objective of CONTEXTGNN into a classification problem with $C$ sampled classes, shared across the entire mini-batch. The utilized sampling procedure is based on a priority queue: we ensure that (1) all ground-truth items and (2) the union of sampled subgraph items of a given mini-batch are included in the set of classes. Then, we (3) fill up the remainder of items with unexplored items outside all subgraphs of the mini-batch based on a uniform sampling procedure. In practice, we have no issues to scale the number of classes $C$ to $\approx 1M$ on commodity GPUs (15GB of memory), giving CONTEXTGNN rich signals to learn from.

## 5 EXPERIMENTAL EVALUATION

We perform experiments on six diverse datasets stemming from different domains, including ten different recommendation tasks. We aim to answer the following research questions:

**Q1** Which benefits does CONTEXTGNN provide over each of its individual component?

**Q2** How does CONTEXTGNN perform against state-of-the-art recommendation system methods?

**Q3** How does the locality score of Sec. 3 influence the model performance of CONTEXTGNN?

**Q4** How efficient and scalable is CONTEXTGNN compared to the related work?

Our method[1] is implemented in ⏻ PYTORCH (Paszke et al., 2019) utilizing the 🔷 PYTORCH GEO-METRIC (Fey & Lenssen, 2019) and 🔲 PYTORCH FRAME (Hu et al., 2024) libraries.

## 5.1 RELATIONAL DEEP LEARNING

**Dataset Description.** We utilize the recommendation tasks introduced in 📊 RELBENCH (Robinson et al., 2024), which consists of eight different realistic and temporal-aware recommendation tasks. 📊 RELBENCH datasets contain rich relational structure, providing a challenging environment for recommendation tasks. The task is to predict the top-$k$ items given a user at a given seed time. The metric we use is Mean Average Precision (MAP) $@ k$, where $k$ is set per task (higher is better).

**`rel-amazon`** Predict the list of distinct items each user purchases (`user-item-purchase`), gives a 5-star (`user-item-rate`), or gives a detailed review (`user-item-review`).

**`rel-hm`** Predict the list of items each user will purchase (`user-item-purchase`).

**`rel-stack`** Predict the list of distinct posts a user will comment (`user-post-comment`), or to predict the list of posts that users will link a given post to (`post-post-related`).

**`rel-trial`** Predict whether a condition will have which sponsors (`condition-sponsor-run`), or whether a sponsor will have a trial in a facility (`site-sponsor-run`).

**Experimental Protocols.** We compare CONTEXTGNN with the following baseline methods:

**LIGHTGBM** (Ke et al., 2017) concatenates both user and item features, and feeds them into a LIGHTGBM decision tree.

**MULTIVAE** (Liang et al., 2018) extends variational autoencoders to collaborative filtering via a user-item interaction matrix for implicit feedback.

**GRAPHSAGE** (Hamilton et al., 2017) employs a heterogeneous GNN on both user and item side, and ranks the produced embeddings via an inner product decoder.

**NGCF** (Wang et al., 2019) extends GRAPHSAGE by propagating both shallow user and item embeddings inside the GNN.

**NBFNET** (Zhu et al., 2021) employs pair-wise GNN representations. This is the backbone of our pair-wise representation model in CONTEXTGNN.

**SHALLOWITEM** describes our two-tower model in CONTEXTGNN, which ranks user GNN representations and shallow item embeddings via an inner product decoder.

Importantly, all GNN-based models utilize the *same* GNN backbone, which guarantees fair comparison of training procedures that are agnostic to the underlying model implementation. Specifically, we use a heterogeneous GRAPHSAGE variant as introduced in Robinson et al. (2024). The hyper-parameters we tune for each task are: (1) the number of hidden units $\in \{32, 64, 128, 256, 512\}$, (2), the batch size $\in \{256, 512, 1024\}$, and (3) the learning rate $\in \{0.001, 0.01\}$.

**Discussion.** The results are reported in Table 2. We answer **Q1** by comparing CONTEXTGNN to its individual components NBFNET and SHALLOWITEM, and answer **Q2** by relating CONTEXTGNN's performance to all reported baselines.

CONTEXTGNN outperforms all competing baselines, often by very significant margins. Notably, one can observe that the two-tower models MULTIVAE, GRAPHSAGE and NGCF all fail to capture the pair-wise signals that both NBFNET and CONTEXTGNN are able to leverage. This shows that two-tower representations are not powerful enough to capture the fine-grained pair-wise dependencies that are required to solve these tasks with high precision. Among the two-tower GNN

---

[1]GitHub: `https://github.com/kumo-ai/ContextGNN`

Table 2: **Recommendation results** (MAP, higher is better, in %) on ⬛ RELBENCH.

| Task | LIGHT GBM | MULTI VAE | GRAPH SAGE | NGCF | NBFNET | SHALLOW ITEM | CONTEXT GNN |
|---|---|---|---|---|---|---|---|
| **rel-amazon** | | | | | | | |
| user-item-purchase | 0.16 | 0.23 | 0.74 | 0.88 | 2.06 | 0.56 | **2.93** |
| user-item-rate | 0.17 | 0.24 | 0.87 | 0.86 | 1.24 | 0.74 | **2.25** |
| user-item-review | 0.09 | 0.10 | 0.47 | 0.55 | 1.57 | 0.40 | **1.63** |
| **rel-hm** | | | | | | | |
| user-item-purchase | 0.38 | 0.28 | 0.80 | 0.75 | 2.81 | 0.40 | **2.93** |
| **rel-stack** | | | | | | | |
| user-post-comment | 0.04 | 0.01 | 0.11 | 0.13 | 12.72 | 0.03 | **13.34** |
| post-post-related | 2.00 | 0.78 | 0.07 | 0.13 | 10.83 | 0.82 | **11.18** |
| **rel-trial** | | | | | | | |
| condition-sponsor-run | 4.82 | 2.47 | 2.89 | 3.88 | 11.36 | 0.85 | **11.65** |
| site-sponsor-run | 8.40 | 6.17 | 10.70 | 6.54 | 19.06 | 10.66 | **28.02** |
| **Average** ($\uparrow$) | 2.01 | 1.29 | 2.08 | 1.72 | 7.71 | 1.81 | **9.23** |

models, there is no clear winner between GRAPHSAGE and NGCF, indicating that shallow (user) embeddings do not significantly drive improvements (and may even hinder performance, especially in tasks with inherent temporal dynamics). CONTEXTGNN improves results by 344% on average compared to the best two-tower baseline.

Among the baselines, NBFNET performs the best across all tasks, while CONTEXTGNN can consistently improve upon these strong outcomes. On average, CONTEXTGNN increases performance by 20% compared to NBFNET, underscoring the importance of incorporating "distant" items into the ranking process - an aspect that NBFNET overlooks by design.

Our own two-tower SHALLOWITEM model performs comparably to other two-tower GNN baselines, despite only using shallow item information in order to improve the overall efficiency of the model. This supports the hypothesis that a deep GNN on the item side is not particularly useful on most tasks, as the shallow embeddings can capture most of the key signals on the item side just as well as the GNN. However, there exists specific cases such as the condition-sponsor-run task on the rel-trial dataset, where SHALLOWITEM underperforms relative to GRAPHSAGE and NGCF. On this task, deep GNNs on the item side play indeed a crucial role to improve results of two-tower models, which is captured in CONTEXTGNN through its pair-wise representation model.

The most noteworthy result is seen in the site-sponsor-run task on the rel-trial dataset. Here, both pair-wise models and two-tower models achieve strong initial results. Combining these two paradigms together via CONTEXTGNN improves the final performance by $\approx 100\%$, indicating that each of the individual components of CONTEXTGNN captures orthogonal signals. The fusion of these paradigms yields a ranking model that excels in both local and distant item ranking, achieving superior overall performance.

In order to answer **Q3**, we now analyze the relationship between the locality score $s_k^{[T,T+i)}$ and the performance of CONTEXTGNN. We observe that the improvements of CONTEXTGNN compared to NBFNET are notably higher on tasks with lower locality scores. This observation aligns intuitively, as CONTEXTGNN needs to rely more heavily on its two-tower component during optimization when locality scores are lower. Specifically, in tasks with the lowest locality scores (*e.g.*, 0.168, 0.170 and 0.229 on the user-item-purchase, user-item-rate and site-sponsor-run tasks from the rel-amazon and rel-trial datasets), CONTEXTGNN achieves substantial performance gains of 42% to 80% compared to NBFNET. Conversely, for tasks with higher locality scores (*e.g.*, 0.417, 0.298, and 0.280 on the user-item-purchase, user-post-comment, and post-post-related tasks from the rel-hm and rel-stack datasets), the performance improvements of CONTEXTGNN over NBFNET are more modest, ranging from 3% to 5%. These findings underscore the significant impact of the locality score on the performance of NBFNET, whereas CONTEXTGNN demonstrates robustness, achieving stellar performance improvements regardless of task-specific characteristics.

## 5.2 STATIC LINK PREDICTION

While CONTEXTGNN's main focus is to excel on large-scale real-world use-cases which are temporal and heterogeneous, it can also be used in a plug-and-play fashion for *any* link prediction task. To verify, we evaluate CONTEXTGNN on the static link prediction task of Amazon-Book (Wang et al., 2019), which is a small-scale dataset of 52,643 users and 91,599 items, which does not come with input features, only considers users with at least ten interactions, and evaluates on 10% of randomly selected interactions independent of time, *cf.* Table 3. We can see that CONTEXTGNN is able to outperform both

Table 3: **Results on Amazon-Book.**

| Model | Recall@20 | NDCG@20 |
|---|---|---|
| NGCF (2019) | 0.0337 | 0.0261 |
| LIGHTGCN (2020) | 0.0410 | 0.0318 |
| ULTRAGCN (2021) | **0.0681** | **0.0556** |
| LIGHTGCL (2023) | 0.0585 | 0.0436 |
| SimGCL (2022) | 0.0478 | 0.0379 |
| SGL (2021) | 0.0468 | 0.0371 |
| **CONTEXTGNN** | 0.0451 | 0.0377 |

NGCF and LIGHTGCN, while it is slightly underperforming compared to, *e.g.*, ULTRAGCN or LIGHTGCL. Given the relatively small size of this dataset, much of the progress in GNN-based recommendation systems has centered on two key directions: (1) simplifying GNN architectures, and (2) incorporating self-supervised learning techniques. Exploring how CONTEXTGNN can benefit from a more lightweight GNN backbone or complementary learning signals presents an exciting direction for future research.

## 5.3 TEMPORAL NEXT-ITEM PREDICTION

We use CONTEXTGNN to perform temporal next-item recommendation on the IJCAI Contest dataset (Xia et al., 2022), which is a common dataset to evaluate sequential recommendation models et al. (Sun et al., 2019; Xia et al., 2020; 2022). As per evaluation protocol, we report HitRate@$k$ and NDCG@$k$ over 99 sampled negatives per entity. Notably, CONTEXTGNN excels at incorporating multi-behavioral and temporal signal,

Table 4: **Results on IJCAI Contest.**

| Model | HR@1 | HR@5 | NDCG@5 | HR@10 | NDCG@10 |
|---|---|---|---|---|---|
| DEEPFM (2017) | 0.138 | 0.332 | 0.244 | 0.469 | 0.290 |
| BERT4REC (2019) | 0.141 | 0.356 | 0.261 | 0.467 | 0.297 |
| CHORUS (2020a) | 0.140 | 0.345 | 0.247 | 0.457 | 0.283 |
| HYREC (2020b) | 0.137 | 0.323 | 0.229 | 0.442 | 0.266 |
| NMTR (2019) | 0.141 | 0.360 | 0.254 | 0.481 | 0.304 |
| MATN (2020) | 0.142 | 0.375 | 0.273 | 0.489 | 0.309 |
| MBGCN (2020) | 0.137 | 0.332 | 0.228 | 0.463 | 0.277 |
| TGT (2022) | 0.148 | 0.399 | 0.293 | 0.519 | 0.330 |
| **CONTEXTGNN** | **0.411** | **0.603** | **0.513** | **0.667** | **0.534** |

*cf.* Table 4. We observe that CONTEXTGNN is able to out-perform all baselines on all metrics on this task (*e.g.*, 170% improvement on HitRate@1).

## 5.4 EFFICIENCY ANALYSIS

Our hybrid CONTEXTGNN model is designed in such a way to have minimal overhead compared to its two components in isolation since the vast majority of the model parts are shared between the two paradigms. To answer **Q4**, we report the runtime in seconds to reach 1,000 optimization steps across different models, *cf.* Table 5. Most importantly, since CONTEXTGNN only requires a single GNN forward pass, it is faster compared to any two-tower GNN by a

Table 5: **Runtime [s]** of 1,000 optimization steps.

| Model | rel-hm | rel-stack |
|---|---|---|
| GRAPHSAGE (1 negative) | 275s | 37s |
| GRAPHSAGE (10 negatives) | 293s | 44s |
| GRAPHSAGE (100 negatives) | OOM | OOM |
| NFBNET | 77s | 21s |
| SHALLOWITEM | 92s | 22s |
| **CONTEXTGNN** | 94s | 23s |

very significant factor (*i.e.*, a two-tower GNN such as GRAPHSAGE requires to run the GNN on both positive and negative items). In particular, two-tower GNNs scale very poorly when increasing the number of negative samples to train against. For training recommendation systems, a large number of negative examples is important to allow learning of discriminative features. However, In the simplest case (one negative pair per positive pair), a two-tower GNN needs to be executed three times already. CONTEXTGNN does not have this limitation and can consider up to 1M negatives before running into GPU memory limitations.

ACKNOWLEDGMENTS

We thank the entire Kumo.AI team for their invaluable support in bringing CONTEXTGNN into production for over dozens of customers and for any dataset scale. Special thanks go to Amitabha Roy, Myungwhan Kim and Federico Reyes Gómez for helpful discussions and feedback.

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
