# OpenReview forum: "ContextGNN: Beyond Two-Tower Recommendation Systems"
_ICLR.cc/2025/Conference — ICLR 2025 Poster_

### Official Review · Reviewer_7mVo · 2024-11-04

**Soundness:** 3
**Presentation:** 4
**Contribution:** 3
**Rating:** 6
**Confidence:** 3

**Summary:**

This paper presents ContextGNN, a refined approach that classifies items into user-specific selections within a user-centric local graph, alongside other items. Subsequently, the GNN learns the pertinent user and item embeddings, thereby enabling the derivation of both pairwise and two-tower representations for enhanced recommendations.

**Strengths:**

1. The paper is well-written and easy to understand.
2. The use of both pairwise and two-tower representations for recommendations is quite interesting.
3. The experiments conducted in this paper seem to demonstrate the model's effectiveness.

**Weaknesses:**

1. The author should consider open-sourcing the code for the proposed method to enhance reproducibility.
2. In the comparison of methods, it would be beneficial to include more recent and advanced GNN-based recommender models beyond NGCF.
3. Given that the author mentions both multi-behavior and temporal graphs, it would be advantageous to illustrate how the GNN backbone is designed to accommodate such graphs.

**Questions:**

See Weaknesses.

---

> ### Author Response · Authors · 2024-11-22
> **Official Answer**
>
> Thank you for your review. Please refer to our general answer which addresses your concerns regarding more advanced baselines beyond NGCF. We have also uploaded our code base to ensure reproducibility of the presented results: https://anonymous.4open.science/r/ContextGNN-DBCA/README.md.
>
> **GNN backbone design:**  In order to disentangle performance improvements of the overarching model architecture from the GNN backbone, we use the same GNN backbone as introduced in RelBench [1], which operates on historical, temporal-aware subgraphs which are sampled on-the-fly during training and evaluation. The GNN backbone takes in raw multi-modal input features across different types (numerical, categorical, multi-categorical, text, absolute time, relative time, *etc*), encodes them into a shared embedding space, and then uses a heterogeneous GNN to perform message passing on top. In particular, we use a heterogeneous version of the GraphSAGE model with sum-based neighbor aggregation.
>
> [1] Robinson et al.: RelBench: A Benchmark for Deep Learning on Relational Databases (2024)

---

### Official Review · Reviewer_U3kG · 2024-11-04

**Soundness:** 2
**Presentation:** 3
**Contribution:** 2
**Rating:** 5
**Confidence:** 3

**Summary:**

* The method uses a pair-wise representation for familiar items in a user's local subgraph and two-tower representations for recommending exploratory items. A final network combines both pair-wise and two-tower recommendations to create a unified ranking of items.

* The study utilizes CONTEXTGNNs in relational deep learning for recommendations within relational databases, which consist of various tables with rich multi-modal features and complex behavioral interactions. This environment provides an ideal setting for testing the model's effectiveness. Additionally, the research explores the challenges of modeling intricate human behavior patterns.

**Strengths:**

* This article has a clear structure and is easy to follow.

*Tested different tasks, which is rare in previous work.

**Weaknesses:**

* It appears that this work has not released the code or dataset, making reproducibility uncertain.

* This work lacks technical innovation and seems quite basic. Such a simple design should emphasize time complexity, runtime efficiency, and scalability, but the paper does not seem to address these aspects.

* This work does not compare with other recent baselines or commonly used models in the industry. Additionally, it does not provide specific data from the dataset. Whether in industry or research, this work does not seem to fit well. Overall, it appears too basic and simple, with relatively little effort involved.

**Questions:**

* The paper mentions the complexity issues of pair-wise methods. Could you provide a time complexity analysis and runtime efficiency experiments of this work compared to others?

---

> ### Author Response · Authors · 2024-11-22
> **Official Answer**
>
> Thank you for your review. Please refer to our general answer for additional baselines/experiments, and the requested efficiency/scalability studies. In summary, we conducted additional experiments on next-item prediction and static link prediction. We find that ContextGNN outperforms sequential recommendation models and is on par to static link prediction models on small-scale datasets. Additionally, we want to highlight that our end-to-end architecture is designed to be scalable and efficient. Our hybrid model is combined in such a way to have minimal overhead compared to the two components in isolation since the vast majority of the model parts are shared between the two paradigms.
>
> We have also uploaded our code base to ensure reproducibility of the presented results: https://anonymous.4open.science/r/ContextGNN-DBCA/README.md. Our code also highlights our technical contributions such as our sampled softmax procedure, the fusion of ranking scores obtained from (approximated) k-NN/MIPS search with ranking scores coming from items in the user’s local subgraph, and the integration of bidirectional sampling in order to ensure that the readout of item nodes is well-defined in a user-centric subgraph.
>
> **Time complexity:** Given that the time complexity of a standard two-tower GNN is $\mathcal{O}(|\mathcal{E}|)$, the total time complexity of our model is $\mathcal{O}(|\mathcal{E}| + |\mathcal{V} \cap \mathcal{R}| + |\mathcal{N}|)$, where $\mathcal{V} \cap \mathcal{R}$ refers to the items within a user’s subgraph and $\mathcal{N}$ refers to the set of negative samples.
>
> In contrast, a standard two-tower GNN needs to be executed on the user, the positive item and the set of negative items $\mathcal{N}$, ending with a much higher time complexity of $\mathcal{O}((2 + |\mathcal{N}|) * |\mathcal{E}|)$. This fundamental difference is also observed in measured runtimes, where ContextGNN is 3 to 5 times faster.

---

### Official Review · Reviewer_pueu · 2024-11-05

**Soundness:** 2
**Presentation:** 2
**Contribution:** 2
**Rating:** 5
**Confidence:** 3

**Summary:**

This paper proposes a recommendation framework that integrates pair-wise representations with two-tower representations. Specifically, it leverages pair-wise representations to capture fine-grained patterns of past user-item interactions, while utilizing two-tower representations for "distant" items. The authors have validated the effectiveness of their approach on the RELBENCH benchmark.

**Strengths:**

1. The motivation behind the proposed method is clear and has significant practical implications for large-scale recommendation systems in the real world.
2. The authors have validated the effectiveness of the proposed method on the benchmark datasets.

**Weaknesses:**

1. The authors emphasize in the abstract and introduction that their method is a relatively general and effective strategy in recommendation scenarios. However, the experimental section only tests the method on datasets that include rich multi-behavioral and temporal interactions. This discrepancy between the claims and the experimental validation raises concerns. Therefore, the authors should either revise their overall statements or consider incorporating a wider variety of datasets for comparative analysis.
2. The experimental section of the paper is quite limited, as it currently only reports the MAP metric and lacks detailed parameter sensitivity analysis. Moreover, the paper does not address how the proposed method's performance may be affected when user behavior information in the dataset is insufficient in type or quantity. Specifically, would we expect a significant decline in performance compared to the baseline algorithms under such conditions? Additionally, I recommend that the authors incorporate a broader range of recent comparison methods to strengthen their empirical evaluation.
3. The paper does not provide the original code, which raises concerns regarding the reproducibility of the results. Additionally, there are some errors in the equations presented.

**Questions:**

Please refer to the weaknesses

---

> ### Author Response · Authors · 2024-11-22
> **Official Answer**
>
> Thank you for your review. Please refer to our general answer which shows the application of ContextGNN in other domains (i.e., static link prediction, next-item recommendation). We have also uploaded our code base to ensure reproducibility of the presented results: https://anonymous.4open.science/r/ContextGNN-DBCA/README.md.  Please note that we only report MAP on RelBench [1] datasets since this is the recommended metric on this benchmark suite. In our additional experimentation, we report other metrics as well (e.g., HitRate@K, Recall@K, NDCG@k).
>
> We have checked all formulas in our paper multiple times, but cannot find any error in these. We kindly ask you to provide the equation which has an error.
>
> **Parameter Sensitivity Analysis:** We found ContextGNN to be very stable across different hyper-parameters, and thus we only tune three parameters in total: number of hidden units, batch size and learning rate. It is important to note that model performance has overall low variance across these different parameters: For example, on the `user-item-purchase` task on `rel-amazon`, the model performance across different hyperparameters ranges from 2.63 to 2.71.
>
> **Sparse User-Behavior:** ContextGNN overcomes sparse user behavior via its two-tower model. In particular, most/all of the recommendations for cold-start users will stem from the two-tower fallback model since the local interaction graph is sparse. This again proves the importance of a hybrid architecture, since cold start users cannot properly be addressed by pair-wise representations, but two-tower models are too general for richer interaction graphs. ContextGNN finds a nice balance to accommodate different types of users into a robust and single architecture.
>
> [1] Robinson et al.: RelBench: A Benchmark for Deep Learning on Relational Databases (2024)

---

> > ### Comment · Reviewer_pueu · 2024-11-26
> >
> > Thank you for the response.

---

### Official Review · Reviewer_uwX2 · 2024-11-05

**Soundness:** 2
**Presentation:** 3
**Contribution:** 2
**Rating:** 5
**Confidence:** 4

**Summary:**

This paper studies the two-tower architecture for recommender systems. They introduce a context-based GNNs-based framework for link prediction in recommendation by using a pair-wise representation technique for similar items in the user's local subgraph. The proposed method achieves promising performance in their experiments.

**Strengths:**

1. The research issue is very important in the industry.

2. Their experimental results demonstrate the effectiveness of the proposed method.

3. The paper is easy to follow.

**Weaknesses:**

However, there are some concerns in the paper:

1. The novelty of the proposed method is unclear.
This paper leverages local interaction graph for representation learning. The idea is very mature and has been well-studied.
 Compared with exciting methods, what are the novelty of the proposed method? Additionally, the related work in the paper can be enhanced.
This paper can be improved by discussing more advanced two-tower architectures. Furthermore, the related work cannot demonstrate the novelty of the proposed method.


2. The technical contributions of this work are also unclear. I was wondering what are the main challenges they are addressing.
The proposed idea looks good. However, the proposed method doesn't have any theoretical analysis.
The model design didn't motivate well.
This paper considers the temporal factor in modeling recommendation. However, what are the technical contributions for taking advantage of it?


3. The experimental section can be enhanced before publication. First, they can compare with some advanced recommendation baselines. There are many advanced recommendation methods for modeling user and item representations.
Second, the datasets here are unclear to me. Whether the proposed method can be used for large-scale datasets?

**Questions:**

See the weakness.

---

> ### Author Response · Authors · 2024-11-22
> **Official Answer**
>
> Thank you for your review. Please refer to our general answer which addresses all your concerns regarding novelty, dataset scale, and comparison to more advanced recommendation baselines. In summary, we conducted additional experiments on next-item prediction and static link prediction. We find that ContextGNN outperforms sequential recommendation models and is on par to static link prediction models on small-scale datasets.
>
> **Related Work:** Thanks for the feedback. We will improve the related work in our revised manuscript. Previously, GNN-based RecSys models have been mostly applied in the static bipartite graph setting (ignoring node features, frequency, seasonality, multi-behavioral interactions, etc) on small-scale datasets only. From NGCF onwards, research has focused on (1) simplifying models (LightGCN [1], Ultra-GCN [2], etc) or by adding self-supervision (e.g., based on contrastive learning via graph augmentations) to improve performance on these small-scale datasets [3, 4, 5]. These findings are orthogonal to our work, but still embrace a two-tower architecture under the hood. The line of work that comes closest to ContextGNN is related to encoding identity- and position-awareness into GNNs in order to relate user-item pairs [6, 7, 8, 9]. However, these methods face scalability challenges as they require a candidate set in advance. ContextGNN builds a trade-off between these two paradigms by bringing scalability and fallback logic into pair-wise embedding models in an elegant way.
>
> **Technical Contributions:** In order to disentangle performance improvements of the overarching model architecture from the GNN backbone, we use the same GNN backbone as introduced in RelBench [10], which operates on historical, temporal-aware subgraphs which are sampled on-the-fly during training and evaluation. The GNN backbone takes in raw multi-modal input features across different types (numerical, categorical, multi-categorical, text, absolute time, relative time, etc), encodes them into a shared embedding space, and then uses a heterogeneous GNN to perform message passing on top. On the training/inference procedure side, technical contributions include the integration of a sampled softmax procedure, the fusion of ranking scores obtained from (approximated) k-NN/MIPS search with ranking scores coming from items in the user’s local subgraph, and the integration of bidirectional sampling in order to ensure that the readout of item nodes is well-defined in a user-centric subgraph. Our entire implementation is written to account for mini-batches of different sizes in a vectorized and GPU-accelerated manner. We have uploaded our code base to also show these technical contributions: https://anonymous.4open.science/r/ContextGNN-DBCA/README.md.
>
> [1] He et al.: LightGCN: Simplifying and Powering Graph Convolution Network for Recommendation (2020)
>
> [2] Mao et al.: UltraGCN: Ultra Simplification of Graph Convolutional Networks for Recommendation (2021)
>
> [3] Yu et al.: Are Graph Augmentations Necessary?: Simple Graph Contrastive Learning for Recommendation (2022)
>
> [4] Wu et al.: Self-supervised Graph Learning for Recommendation (2021)
>
> [5] Cai et al.: LightGCL: Simple Yet Effective Graph Contrastive Learning for Recommendation (2023)
>
> [6] You et al.: Identity-aware Graph Neural Networks (2021)
>
> [7] You et al.: Position-aware Graph Neural Networks (2019)
>
> [8] Zhu et al.: Neural Bellman-Ford Networks: A General Graph Neural Network Framework for Link Prediction (2021)
>
> [9] Zhang et al.: Link Prediction Based on Graph Neural Networks (2018)
>
> [10] Robinson et al.: RelBench: A Benchmark for Deep Learning on Relational Databases (2024)

---

> ### Comment · Reviewer_uwX2 · 2024-12-03
>
> Thank you for the response.

---

### Official Review · Reviewer_GUBd · 2024-11-05

**Soundness:** 3
**Presentation:** 3
**Contribution:** 3
**Rating:** 8
**Confidence:** 4

**Summary:**

The paper proposes ContextGNNs, a new architecture for graph-based recommendation. Unlike previous similar approaches, such as pair-wise recommendation solutions, the authors propose a pair-wise representation only for exploring the items in the user's local subgraph, while the architecture employs the usual two-tower method for the item side. Finally, a network is designed and trained to fuse the two representations and provide the final recommendations.

On the one hand, the ContextGNNs framework first generates a pair-wise representation for users and items, where a $k$-hop subgraph for each user is sampled, and a GNN is leveraged to obtain embedding representations and the overall item ranking. On the other hand, a two-tower representation is exploited on the item side, as it can be shown that: 1) GNNs do not learn useful information on the item side as the item interaction matrix tends to be very dense; 2) shallow item embeddings are effective; 3) GNN item representations can scale poorly with large numbers of items in the catalog. The so-learned item embeddings are injected into the GNN forward pass presented above, to allow the user representations to align with the item ones. Finally, an MLP is trained to fuse the two designed contributions.

Results on different tasks and datasets from the RelBench platform demonstrate the efficacy of the ContextGNNs proposed baseline, also when compared to different settings of the same model.

**Strengths:**

\+ The proposed approach is well-placed within the existing literature on (graph)-based recommendation, and the motivations behind ContextGNNs are quite solid and sound

\+ Overall, the paper is well-written and easy to follow

\+ I appreciated the locality score study, which conceptually and empirically supports the rationale behind the approach

\+ I also appreciated how the authors discussed possible extensions of the proposed approach in different and interesting new directions

\+ Even considering the page limitations, I believe the experimental setting is adequately extensive to empirically support the rationale behind the paper

**Weaknesses:**

\- To my understanding, it seems that the list of baselines for the experiments is not completely diversified and updated to the latest advances in graph-based recommendation. I would have appreciated seeing more recent and popular approaches being tested against ContextGNNs, such as LightGCN (cited in the paper), SGL [i], UltraGCN [ii], SimGCL [iii], and LightGCL [iv].

**References**

[i] Jiancan Wu, Xiang Wang, Fuli Feng, Xiangnan He, Liang Chen, Jianxun Lian, Xing Xie: Self-supervised Graph Learning for Recommendation. SIGIR 2021: 726-735

[ii] Kelong Mao, Jieming Zhu, Xi Xiao, Biao Lu, Zhaowei Wang, Xiuqiang He: UltraGCN: Ultra Simplification of Graph Convolutional Networks for Recommendation. CIKM 2021: 1253-1262

[iii] Junliang Yu, Hongzhi Yin, Xin Xia, Tong Chen, Lizhen Cui, Quoc Viet Hung Nguyen: Are Graph Augmentations Necessary?: Simple Graph Contrastive Learning for Recommendation. SIGIR 2022: 1294-1303

[iv] Xuheng Cai, Chao Huang, Lianghao Xia, Xubin Ren: LightGCL: Simple Yet Effective Graph Contrastive Learning for Recommendation. ICLR 2023

**Questions:**

I have a question that does not refer to the one weakness raised. As observed by the authors in the locality score study, it is interesting to see that many validation/test links are not in the users' nearest neighborhood. Moreover, we know from the literature, that graph-based recommender systems may be affected by oversmoothing. On such a basis, my question is: how the graph-based model can learn long-range (and, evidently, ground-truth) products connected to users if the model cannot explore on more than ~3 hops (on average)? Do you have any intuitions for that?

---

> ### Author Response · Authors · 2024-11-22
> **Official Answer**
>
> Thank you very much for your review and your positive feedback. Please refer to our general answer for additional baselines/experiments, and efficiency/scalability studies.
>
> As you correctly pointed out, the locality study reveals that a vast majority of ground-truth items are not part of a user’s local subgraph, and hence are not aggregated into its user representation. Still, the user representation holds valuable information about past interactions and collaborative signals. However, ContextGNN falls back to the two-tower model in order to compute a ranking score for these “distant” items. The key idea is that the model learns to give more weight to the two-tower part in case such long-range/exploratory interactions frequently happen in the dataset, either on a per-user basis or on a global level. In practice, it is often not feasible to leverage more than 3-hops when using subgraph sampling approaches, and deeper GNNs are limited to full-batch processing on smaller-scale datasets. We still use recent tricks such as skip-connections to mitigate the effects of oversmoothing in our GNN backbone.

---

> > ### Comment · Reviewer_GUBd · 2024-11-22
> > **Response to authors' rebuttal**
> >
> > Dear Authors,
> >
> > Thank you for your detailed response to all reviewers' concerns, which I carefully read.
> >
> > Regarding my (minor) concern, I think you addressed it well. On the one hand, it might seem that, compared to other more novel and advanced GNN-based approaches, your model cannot (always) be competitive. However, it's also true that the rationales behind your work were (and are) different, at least from what I understood. Indeed, I think your work was not intended to be yet another GNN-based recommender system beating the most recent state-of-the-art by a large margin, but rather to provide another perspective on the field. In this respect, I strongly believe new research papers should not only be about outperforming the state-of-the-art if there's not much logic behind it.
> >
> > Moreover, you're right when you say that most of the models I suggested to compare as baselines would not be scalable in a real-world/industrial setting.
> >
> > Still, I'd personally be interested in seeing how more recent solutions (such as contrastive learning) may work on your approach, if this may cause scalability problems, etc... But I guess this line would be great for future work.
> >
> > Thank you also for your opinion regarding my question (a curiosity).
> >
> > I don't have any other concerns regarding the paper. I won't change my initial score as the weakness I pointed out was not affecting it that much. I still believe the score I gave is suitable for the work at this stage.
> >
> > Best of luck with the other reviews!

---

### Author Response · Authors · 2024-11-22
**Answer to all Reviewers [1/3]**

We thank all reviewers for their constructive feedback and helpful suggestions. We are glad that they found our work interesting (**7mVo**), well motivated (**GUBd**, **pueu**), important (**uwX2**), well-supported by experiments (**GUBd**, **pueu**, **7mVo**), easy-to-follow (**GUBd**, **7mVo**, **uwX2**, **U3kG**), and well-placed in literature (**GUBd**). In this response, we address joint comments of the reviewers while also providing individual responses to each reviewer separately. We hope that reviewers’ concerns are addressed well and would kindly ask for a score increase if that is the case or for further suggestions on how to further improve the work.

**Code Submission:** Based on reviewers’ requests, we have uploaded our code base to ensure reproducibility of the presented results: https://anonymous.4open.science/r/ContextGNN-DBCA/README.md

**Contribution and Novelty:** Reviewers **uwX2** and **U3kG** had concerns about the novelty of our work. The novelty of our model is the fusion of both pair-wise and two-tower-based recommendations into a single architecture. Since most GNN-based RecSys models are two-tower based (e.g., NGCF [1], LightGCN [2], UltraGCN [3], SimGCL [4], SGL [5], LightGCL [6], etc), they inherit the shared fundamental weakness that the item representations are not capturing the uniqueness of a user’s view on the items since users and items are ranked based on two completely separate, pair-agnostic representations. On the other hand, pair-wise representations (which captures the user-specific view on items) can not be computed for all possible user-item pairs due to its quadratic complexity. Our contribution is a novel end-to-end mixture of these two paradigms, which is efficient, scalable, and improves performance out-of-the-box. This fundamentally changes how we think about GNN-based recommendation systems since ContextGNN integrates signals into the model that are unable to be captured by a two-tower model (e.g., repeated purchases, items that have been previously looked at, friends who just bought this particular item, co-purchases, etc), while preserving the strengths of two-tower models for exploratory purchases (e.g., popularity bias, content-based filtering, etc). Our efficiency study below also showcases that ContextGNN not only improves performance, it is also significantly faster than two-tower based approaches.

The importance of our contribution is supported further by our study on real-world datasets (Section 3), which reveals that user behavior is very diverse: Some users prefer to interact with “familiar” items local to the user, while others like to constantly explore new items distant to the user. This shows the necessity of introducing a hybrid architecture.

---

> ### Author Response · Authors · 2024-11-22
> **Answer to all Reviewers [2/3]**
>
> **Additional Experiments:** Reviewers requested additional experiments with alternative baselines (**GUBd**, **uwX2**, **U3kG** and **7mVo**) and alternative datasets (**pueu**). While we want to point out that we already evaluate ContextGNN on 8 different tasks, and it is non-trivial to transfer related work to the RelBench [8] setting, we agree that this is a great suggestion. As such, we conducted two more lines of experiments to further strengthen our experimental evaluation:
>
> We use ContextGNN to perform temporal next-item recommendation on the ICJAI Contest dataset [7], which is a common dataset utilized to evaluate sequential recommendation models (Bert4Rec, TGT et al.). As per evaluation protocol, we report HitRate@k and NDCG@k over 99 sampled negatives per entity (random HitRate@10 performance =0.1):
>
> ```
> | Model      |  HR@1 |  HR@5 | NDCG@5 | HR@10 | NDCG@10 |
> | DeepFM     | 0.138 | 0.332 |  0.244 | 0.469 |   0.290 |
> | Bert4Rec   | 0.141 | 0.356 |  0.261 | 0.467 |   0.297 |
> | Chorus     | 0.140 | 0.345 |  0.247 | 0.457 |   0.283 |
> | HyRec      | 0.137 | 0.323 |  0.229 | 0.442 |   0.266 |
> | NMTR       | 0.141 | 0.360 |  0.254 | 0.481 |   0.304 |
> | MATN       | 0.142 | 0.375 |  0.273 | 0.489 |   0.309 |
> | MBGCN      | 0.137 | 0.332 |  0.228 | 0.463 |   0.277 |
> | TGT        | 0.148 | 0.399 |  0.293 | 0.519 |   0.330 |
> | ContextGNN | 0.411 | 0.603 |  0.513 | 0.667 |   0.534 |
> ```
>
> We observe that ContextGNN is able to out-perform all baselines on all metrics on this task (e.g., ~170% improvement on HitRate@1).
>
> Reviewers also requested comparison to GNN-based RecSys models such as LightGCN, Ultra-GCN, etc.  Although our main focus in ContextGNN is on mid to large-scale real-world use-cases which are temporal and heterogeneous, we additionally evaluate ContextGNN on the well-known static link prediction task in Amazon-Books, which is a small-scale dataset that does not come with input features, only considers users with at least ten interactions, and evaluates on 10% randomly selected interactions independent of time.
>
> ```
> | Model      | Recall@20 | NDCG@20 |
> | NGCF       |    0.0337 |  0.0261 |
> | LightGCN   |    0.0410 |  0.0318 |
> | Ultra-GCN  |    0.0681 |  0.0556 |
> | LightGCL   |    0.0585 |  0.0436 |
> | SimGCL     |    0.0478 |  0.0379 |
> | SGL        |    0.0468 |  0.0371 |
> | ContextGNN |    0.0451 |  0.0377 |
> ```
>
> We can see that ContextGNN is able to outperform both NGCF and LightGCN, while it is slightly underperforming compared to, *e.g.*, Ultra-GCN or LightGCL. Given the relatively small size of this dataset, much of the progress in GNN-based recommendation systems has centered on two key directions: (1) simplifying GNN architectures, as seen in the progression from NGCF to LightGCN and Ultra-GCN, and (2) incorporating self-supervised learning techniques, such as SimGCL, SGL, and LightGCL, to enhance the learning signal. Note that we did not make any modifications to our GraphSAGE-like GNN backbone nor do we introduce complex self-supervision pipelines, yet we achieve superior performance compared to both NGCF and LightGCN on this task. Exploring how ContextGNN can benefit from a more lightweight GNN backbone or complementary learning signals, akin to those in SimGCL, SGL, or LightGCL, presents an exciting direction for future research.

---

> ### Author Response · Authors · 2024-11-22
> **Answer to all Reviewers [3/3]**
>
> **Efficiency:** Reviewer **U3kG** requested an additional efficiency study. We want to highlight that our end-to-end architecture is designed with scalability and efficiency in mind. Our hybrid model is combined in such a way to have minimal overhead compared to the two components in isolation since the vast majority of the model parts are shared between the two paradigms. To verify, we report the runtime in seconds to reach 1000 optimization steps across different models:
>
> ```
> | Model                        | rel-hm | rel-stack |
> | GraphSAGE (1 negative)       |   275s |       37s |
> | GraphSAGE (10 negatives)     |   293s |       44s |
> | GraphSAGE (100 negatives)    |    OOM |       OOM |
> | Pair-wise Part (NBFNet)      |    77s |       21s |
> | Two-tower Part (ShallowItem) |    92s |       22s |
> | ContextGNN (200k negatives)  |    94s |       23s |
> ```
>
> Most importantly, since ContextGNN only requires user GNN embeddings in its two-tower part, it is faster compared to any two-tower GNN by a very significant factor (i.e., a two-tower GNN such as GraphSAGE requires to run the GNN on both positive and negative items). In particular, two-tower GNNs scale very poorly when increasing the number of negative samples to train against. For training recommendation systems, a large number of negative examples is important to allow learning of discriminative features (see below for a quantitative analysis). However, In the simplest case (one negative pair per positive pair), a two-tower GNN needs to be executed three times already. ContextGNN does not have this limitation and can consider up to 1M negatives before running into GPU memory limitations.
>
> **Scalability:** Reviewer **U3kG** requested an additional scalability study. ContextGNN is very scalable and is evaluated across a diverse range from medium to large-scale datasets, which outpass commonly used datasets such as Gowalla (29k users/40k items), Yelp2018 (31k users/38k items), and Amazon-Book (52k users/91k items) by a large factor:
>
> ```
> |   Dataset  | #Nodes |
> | rel-amazon |    15M |
> | rel-avito  |    21M |
> | rel-event  |    41M |
> | rel-hm     |    16M |
> | rel-stack  |     4M |
> | rel-trial  |     5M |
> ```
>
> In order to scale the shallow embedding part of ContextGNN, we make use of a sampled softmax formulation, which allows us to scale to up to 1M negative samples on commodity GPUs (15GB of memory). For this, we report model performance and runtime when sweeping over the number of negatives:
>
> ```
> | Number of negatives |      rel-hm     | rel-trial (sponsor) |
> |                     |  MAP  | Runtime |    MAP   |  Runtime |
> |                  10 |  1.36 |     77s |     18.0 |     421s |
> |                 100 |  2.00 |     77s |     19.6 |     423s |
> |               1,000 |  2.35 |     78s |     22.4 |     425s |
> |              10,000 |  2.70 |     82s |     26.8 |     426s |
> |             100,000 |  2.93 |     93s |     26.9 |     427s |
> ```
>
> We found that increasing the corpus of negative samples increases final model performance at the expense of a small bit of runtime. This intuitively makes sense, as just a small number of uniformly sampled negatives may not sufficiently differentiate between positive examples and challenging negative counterparts. In practice, we use the largest number of negative samples possible without running into GPU memory limitations.
>
> [1] Wang et al.: Neural Graph Collaborative Filtering (2020)
>
> [2] He et al.: LightGCN: Simplifying and Powering Graph Convolution Network for Recommendation (2020)
>
> [3] Mao et al.: UltraGCN: Ultra Simplification of Graph Convolutional Networks for Recommendation (2021)
>
> [4] Yu et al.: Are Graph Augmentations Necessary?: Simple Graph Contrastive Learning for Recommendation (2022)
>
> [5] Wu et al.: Self-supervised Graph Learning for Recommendation (2021)
>
> [6] Cai et al.: LightGCL: Simple Yet Effective Graph Contrastive Learning for Recommendation (2023)
>
> [7] Xia et al.: Multi-Behavior Sequential Recommendation with Temporal Graph Transformer (2022)
>
> [8] Robinson et al.: RelBench: A Benchmark for Deep Learning on Relational Databases (2024)

---

> > ### Comment · Reviewer_GUBd · 2024-11-22
> > **Response to authors' rebuttal**
> >
> > Dear Authors,
> >
> > Thank you for your detailed answers to all reviews. I think you addressed the vast majority of concerns raised by all the reviewers.
> >
> > Please refer below to my answer to the rebuttal for detailed explanations.

---

### Author Response · Authors · 2024-11-27
**Revised manuscript**

Dear reviewers,

we also revised our manuscript and have uploaded a new version. All changes to the initial version are marked in blue.

---

> ### Comment · Reviewer_GUBd · 2024-11-27
>
> Thank you very much!

---

### Meta-Review · Area_Chair_6N9T · 2024-12-21

**Metareview:**

This paper presents a novel recommendation framework that combines pair-wise representations with two-tower representations to enhance recommendation performance. The framework employs pair-wise representations to capture fine-grained patterns from past user-item interactions, while leveraging two-tower representations to handle "distant" items. The effectiveness of this approach is demonstrated through extensive validation on the RELBENCH benchmark.

Positive points:
+ The paper is well written
+ The idea proposed in the paper is sound and interesting
+ The experiments are very extensive, which can well support the authors' claims

Negative points:
- The baselines are not sufficiently up-to-date.
- The original code is not provided in the paper

**Additional Comments On Reviewer Discussion:**

In the rebuttal period, the authors have provided very detailed responses. Reviewer GUBd strongly supports the paper in the discussion period. I have also carefully read the comments and the authors' feedback, I agree with reviewer GUBd, and would like to accept this paper.

---

### Decision · Program_Chairs · 2025-01-22

Accept (Poster)